# Electrospun Poly(L-lactide-co-ε-caprolactone) Scaffold Potentiates C2C12 Myoblast Bioactivity and Acts as a Stimulus for Cell Commitment in Skeletal Muscle Myogenesis

**DOI:** 10.3390/bioengineering10020239

**Published:** 2023-02-11

**Authors:** Serafina Pacilio, Roberta Costa, Valentina Papa, Maria Teresa Rodia, Carlo Gotti, Giorgia Pagnotta, Giovanna Cenacchi, Maria Letizia Focarete

**Affiliations:** 1Department of Biomedical and Neuromotor Sciences DIBINEM, Alma Mater Studiorum—University of Bologna, 40100 Bologna, Italy; 2Applied Biomedical Research Center—CRBA, IRCCS St. Orsola Hospital, Alma Mater Studiorum—University of Bologna, 40100 Bologna, Italy; 3Department of Chemistry “Giacomo Ciamician”, INSTM UdR of Bologna, University of Bologna, 40100 Bologna, Italy; 4Interdepartmental Center for Industrial Research in Advanced Mechanics and Materials (CIRI-MAM), Alma Mater Studiorum—University of Bologna, 40100 Bologna, Italy; 5Health Sciences & Technologies (HST) CIRI, University of Bologna, Via Tolara di Sopra 41/E, 40064 Ozzano Emilia, Italy

**Keywords:** skeletal muscle, myogenesis, biomaterials, polylactide copolymers, electrospinning, tissue engineering, 3D cell culture

## Abstract

Tissue engineering combines a scaffold, cells and regulatory signals, reproducing a biomimetic extracellular matrix capable of supporting cell attachment and proliferation. We examined the role of an electrospun scaffold made of a biocompatible polymer during the myogenesis of skeletal muscle (SKM) as an alternative approach to tissue regeneration. The engineered nanostructure was obtained by electrospinning poly(L-lactide-co-ε-caprolactone) (PLCL) in the form of a 3D porous nanofibrous scaffold further coated with collagen. C2C12 were cultured on the PLCL scaffold, and cell morphology and differentiation pathways were thoroughly investigated. The functionalized PLCL scaffold recreated the SKM nanostructure and performed its biological functions, guiding myoblast morphogenesis and promoting cell differentiation until tissue formation. The scaffold enabled cell–cell interactions through the development of cellular adhesions that were fundamental during myoblast fusion and myotube formation. Expression of myogenic regulatory markers and muscle-specific proteins at different stages of myogenesis suggested that the PLCL scaffold enhanced myoblast differentiation within a shorter time frame. The functionalized PLCL scaffold impacts myoblast bioactivity and acts as a stimulus for cell commitment, surpassing traditional 2D cell culture techniques. We developed a screening model for tissue development and a device for tissue restoration.

## 1. Introduction

Frequent reasons for skeletal muscle (SKM) loss are injuries, surgery, ageing, metabolic diseases and inherited genetic diseases. Although SKM wasting is partially restored by satellite cells (SCs), a population of myogenic precursor cells located between the basal lamina and sarcolemma of mature myofibers [1], it has been demonstrated that resident SCs lose their self-regenerative potential with age and disease [2]. As a result of the lack of regenerative capacity and surgical interventions and limited therapeutic options, SKM loss is considered largely irreversible [3,4]. New strategies are required to overcome the limitations of conventional therapies for muscle regeneration [5,6,7], and to date, many attempts have been made to engineer SKM tissue in vitro with synthetic polymeric biomaterials. An in vitro model is a simple and realistic method widely used in medicine and biology to study cells and tissues and to perform functional tests. Although tissue culture plastics (TCPs) are the most commonly used support for 2D in vitro cell culture, their stiffness and bidimensional structure make them unsuitable for advanced in vitro studies. Extensive research has been conducted with the goal of fabricating functional 3D supports, known as scaffolds, that represent the tissue microenvironment more realistically and with a structure modulable according to experimental needs [8,9]. The scaffold, together with regulatory signals, provides a biomimetic extracellular matrix (ECM) for cell growth and proliferation and represents a key element in tissue engineering (TE) [10,11,12]. The electrospinning (ESP) technique has been used to fabricate electrospun scaffolds for many TE strategies [13]. In particular, synthetic aliphatic polyesters have been successfully electrospun because of their mechanical properties, commercial availability and Food and Drug Administration (FDA) approval [14]. Using ESP, it is possible to obtain fibers with controlled dimensions and orientation, providing an anisotropic structural organization of SKM tissue [15,16]. Indeed, SKM development begins with myoblasts derived from myogenic precursors and ends with the formation of multinucleated myotubes following myocyte fusion; the final in vivo product consists of a mature muscle fiber composed of parallel myofibrils [17], each with sequentially aligned sarcomeres [18]. In this regard, the electrospun scaffold can guide myofibrils’ alignment and support SKM tissue formation, which is not allowed with traditional TCP [19].

Several studies have been conducted to test the usefulness (advantages and disadvantages) of different electrospun scaffolds for SKM tissue fabrication [20]. For example, Choi et al. [21], using an electrospun aligned poly(3-caprolactone) (PCL)/collagen nanofiber scaffold, showed that the presence of a topographical cue was only necessary for myoblast alignment, whereas an elastic substrate was required for myotube differentiation. This idea was elaborated by Krishna H Patel [22], who highlighted the need for a micropattern scaffold capable of guiding cells into organized structures and stimulating their proliferation using a mixture of PCL and a decellularized ECM. An interesting work by L. Wang et al. [23] elucidated the importance of integrating an electrospun scaffold with additional biomaterial; a scaffold of aligned nanofibers and hydrogels allowed the production of parallel and elongated myotubes. Since in their native environment, cells are surrounded by different structures with different scales, it is not surprising that cell attachment, proliferation and differentiation could be affected by the presence of external micropatterned material [24]. However, external stimuli must be constant in order to have an impact on cells; consequently, a scaffold must maintain its mechanical integrity. It is well-known that copolymerization of lactide (LA) with the flexible caprolactone (CL) unit is an appropriate method for controlling the mechanical properties and degradation rate of the resulting materials [25,26], widening the range of biomedical applications of this family of polymers and making them ideal materials for applying steady stimuli in vitro or in vivo [27]. Accordingly, in the present work, we designed an electrospun scaffold made of poly(L-lactide-co-ε-caprolactone) (PLCL) further coated with type I collagen to recreate a favorable environment for cells which, in recognizing the collagen as the main constituent of the ECM of SKM, succeed in adhering to the substrate [28]. The scaffold is characterized by controlled microarchitecture to promote cell growth, by suitable physiochemical properties to favour cell interactions, by appropriate mechanical properties to guarantee cell functionality and by a customized morphology to resemble the native environment, all of which, taken together, allowed us to further study the phenotypic profile of myoblasts throughout SKM tissue formation. As suggested by E. Martinez [29] and Huaqiong Li [24], micro- and nanostructure can lead to deformation of cells and, reflexively, the nucleus, which is attached to the cytoskeleton by intermediate filaments. Thus, we explored the effects of a polymer-based scaffold on the differentiation pathway at different stages of myogenesis using accurate quantitative analysis of differentiation-specific markers, examining myogenic regulatory factors (MRFs) that directly control the myogenic process and interact with myocyte-specific enhancer factors (MEFs), which regulate myoblast commitment [30,31]. Furthermore, myoblast morphology towards myogenic commitment was studied under both confocal and electron microscopy to determine the role of the scaffold in shaping SKM myogenesis and tissue growth by detecting the interactions that modulate myoblast, myocyte and myotube formation, thereby unravelling the link between cell fate and material cues. The findings identify pivotal events in SKM development that are indispensable for either obtaining a clearer explanation of physiological tissue formation or for recognizing unusual tissue maturation predictive of dysfunctional or pathological SKM.

## 2. Materials and Methods

### 2.1. Materials

Poly(L-lactide-co-Ɛ-caprolactone) (PLCL) copolymers with different compositions were purchased from Corbion (Amsterdam, The Netherlands). In particular, PLCL 70/30 (LA/CL molar ratio) (PURASORB^®^ PLC 7015, inherent viscosity midpoint of 1.5 dL/g), PLCL 85/15 (PURASORB^®^ PLC 8516, inherent viscosity midpoint of 1.6 dL/g) and PLCL 95/15 (PURASORB^®^ PLC 9517, inherent viscosity midpoint of 1.7 dL/g) were used. Dichloromethane (DCM) and dimethylformamide (DMF) were supplied by Sigma-Aldrich and used without further purification. Type I collagen solution from calf skin (acetic acid 0.25%) was obtained from Sigma-Aldrich (St. Louis, MO, USA).

### 2.2. Scaffold Fabrication

The homemade electrospinning apparatus consisted of a high-voltage power supply (Spellman SL 50 P 10/CE/230), a syringe pump (KD Scientific 200 series) and a glass syringe containing the polymer solution connected to a stainless steel blunt-ended needle. A rotating collector (metallic drum: length = 120 mm, diameter = 50 mm) was used to produce electrospun scaffolds made of uniaxially aligned nanofibers. Electrospinning was carried out at room temperature (RT) and a relative humidity of 50%. Polymer solutions were obtained by dissolving each of the three PLCL samples in a mixture of DCM:DMF = 65:35 *v*/*v* at a polymer concentration of 20% *w*/*v*. The polymeric solution was delivered at a constant flow rate (1.2 mL/h), with a needle-to-collector distance of 20 cm and a voltage of 20 kV. To obtain aligned nanofibers, the collector was set in rotation at a linear speed of 15.7 m/s, whereas random fibers were obtained using a linear speed of 6000 rpm. After a 90 min long process, a 30–40 μm thick, 15 × 8 cm electrospun mat was obtained.

### 2.3. Scaffold Characterization Methods

Scanning electron microscopic (SEM) observations were performed using a Phenom Pro-X SEM (Thermo Fisher Scientific Inc., Waltham, MA, USA) at an accelerating voltage of 10 kV on gold-sputtered samples. The SEM images were analyzed using Fiji, an open-source software based on ImageJ2 [32], to measure the distribution of the diameter of the nanofibers; results are reported as the average diameter ± standard deviation (SD). The fiber orientation of the mats of aligned nanofibers was calculated using the Directionality plugin of Fiji [32], employing the local gradient orientation method following a previously validated procedure [33]. Thermal transitions were measured by means of a differential scanning calorimeter (DSC Q100; TA Instruments, New Castle, DE, USA). About 5 mg of sample was placed in Tzero aluminum pans and subjected to a heating scan at 20 °C/min from −80 °C to 200 °C, quenched to −80 °C and heated up to 200 °C at 20 °C/min under a nitrogen atmosphere. Stress–strain measurements were performed with an Instron 4465 tensile testing machine on rectangular strips (width = 5 mm, gauge length = 20 mm) cut from the electrospun mats, with the gauge length aligned with the nanofiber direction. The crosshead speed was set to 10 mm/min, corresponding to a strain rate of 50%/min. Specimens and results are reported as the average value ± SD and converted to stress–strain curves.

### 2.4. Cell Seeding Conditions

Prior to cell seeding, all scaffolds were cut into suitably sized pieces and assembled with CellCrown™ support (Scaffdex, Tampere, Finland), then sterilized using ethanol following a previously described protocol [34].

### 2.5. Scaffold Functionalization

Type I collagen solution from calf skin was diluted 10-fold with distilled water for a working concentration of 0.01% and used to coat sterilized scaffolds; coating was applied on the scaffolds overnight at 4 °C. The coating solution was removed, and scaffolds were washed with PBS before cell seeding. FT-IR was carried out using a Spectrum Two instrument equipped with an ATR accessory (Perkin-Elmer, diamond crystal, Milan, Italy) on both uncoated and coated scaffolds. All spectra were registered between 400 cm^−1^ and 4000 cm^−1^ with a resolution of 4 cm^−1^, an accumulation of 16 scans and a step size of 1 cm^−1^. Water contact angle (WCA) was assessed by the sessile drop method in air with a time of analysis of 10 s using a KSV CAM101 instrument (KSV Instruments Ltd., Helsinki, Finland) on uncoated and coated scaffolds. Ten measurements were performed for each sample.

### 2.6. Cell Culture

C2C12 murine myoblasts (ATCC Cat# CRL-1772, RRID:CVCL 0188) were plated into a complete growth medium composed of Dulbecco’s modified Eagle high-glucose medium (DMEM) with 10% fetal bovine serum (FBS) (Biowest, Nuaille, France), L-glutamine (Euroclone, Milan, Italy) and 1% penicillin/streptomycin at 37 °C and 5% CO_2_. At 80% confluence, C2C12 were induced to differentiate, replacing complete growth medium with a differentiation medium made of DMEM with 1% horse serum (HS) (Sigma-Aldrich, St. Louis, MO, USA). Myogenic differentiation was investigated as follows: T0, proliferating undifferentiated myoblast used as control; T1, early stage after 24 h of differentiation; T3–T5, intermediate stage after 3–5 days of differentiation; T7–T10, late stage. C2C12 were cultured with the same cell-seeding density (15 × 10^3^ cells/cm^2^) under two conditions: standard TCP as a control and a non-woven PLCL scaffold.

### 2.7. Cell Viability Assay

C2C12 were seeded on the TCP and on the PLCL scaffold, and cell viability and proliferation were determined using the Cell Titer 96 AQueous One Solution Cell Proliferation Assay (Promega, Madison, WI, USA) according to the manufacturer’s instructions. The absorbance value was read at 490 nm using a Spark Tecan microplate reader and tests assessed at four time points (days 1, 2, 5 and 7), each performed in triplicate.

### 2.8. Morphological Study

The morphology of the myogenesis evolution of cells cultured on the PLCL scaffolds was observed with SEM after 2, 6 and 24 h of culture. Cell-laden PLCL scaffolds were fixed in 2.5% glutaraldehyde (TAAB Laboratories) in sodium cacodylate 0.1 M (Electron Microscopy Sciences) at pH 7.2, for 3 h at 4 °C, washed twice in sodium cacodylate buffer 0.15 M at RT and fixed in osmium tetroxide 1% (OsO4) in cacodylate buffer 0.1 M for 30′ at 4 °C. Next, samples were washed once in cacodylate buffer 0.1 M for 10 min at RT, dehydrated with ascending ethanol solutions (70%, 95% and 100%) and a 1:1 solution of 100% ethanol and hexamethyldisilazide for 1 h at RM and left overnight in hexamethyldisilazide at RT. Cell-seeded PLCL scaffolds were processed for SEM analysis following the procedure described in Scaffold Characterization Methods.

### 2.9. Quantitative Reverse Transcription Polymerase Chain Reaction (RT-qPCR)

RNA of the C2C12 control on TCP and PLCL scaffold at T0 and from T1 to T7 was extracted using TRIZOL^®^ reagent (Thermo Fisher Scientific, Waltham, MA, USA) as previously published [35]. One microgram of RNA was reverse-transcribed, and real-time qPCR was performed with MaximaTM SYBR Green qPCR Master Mix 2X (both kits from Thermo Fisher Scientific) in an IQ5 Thermal Cycler RT-PCR detection system (BioRad, Hercules, CA, USA). Real-time qPCR analysis was performed in triplicate, and qPCR signals (CT) were normalized to glyceraldehyde 3-phosphate dehydrogenase (GAPDH). Primers are listed in Table 1.

### 2.10. Western Blotting

The C2C12 controls on TCP and PLCL scaffolds were lysed with RIPA buffer at T0 and from T1 to T10 as described by Costa et al. [35]; then, lysates were collected and stored at −80 °C. Individual protein concentration measurements were performed using a Lowry assay kit. Proteins were separated on Invitrogen NuPage mini gels (Thermo Fisher Scientific, Waltham, MA, USA), transferred to a nitrocellulose membrane and blocked in 5% skimmed milk dissolved in TBS-Tween (0.1% Tween) for 1 h at RT. Membranes were incubated overnight at 4 °C with primary antibody antiskeletal muscle myosin (F59) (dilution 1:200; Santa Cruz Biotechnology, Dallas, TX, USA) and incubated for 1 h at RT with peroxidase-labeled secondary antibody. After washing, membranes were exposed for 1 minute to ECL Western blotting substrates (1:1), and chemiluminescent signals were detected (Clarity Western ECL Substrate, BioRad Hercules, CA, USA). The relative intensity of protein expression was calculated using ImageJ2 and normalized to actin.

### 2.11. Immunostaining

For immunofluorescence (IF), cell-laden PLCL scaffolds were fixed in PFA 4% for 20 min and later washed three times in PBS. After incubation with primary antibody antiskeletal muscle myosin (F59) (dilution, 1:200; Santa Cruz Biotechnology, Dallas, TX, USA) overnight at 4 °C, cells were washed, incubated with a specific secondary antibody (Goat Anti-Mouse IgG (H+L), DyLight 488; dilution, 1:1000; Thermo Fisher Scientific, Waltham, MA, USA) for 1 h at 37 °C, and nuclei were counterstained with Hoechst (Sigma-Aldrich, St. Louis, MO, USA). Slides were mounted with aqueous medium. Confocal imaging was performed using a Nikon A1 confocal laser scanning microscope equipped with a 40×, 1.4 NA objective and with 405 and 488 nm laser lines. Z-stacks were collected at an optical resolution of 210 nm/pixel, stored at 12-bit with 4096 different gray levels and the pinhole diameter set to 1 Airy unit and z-step size were set to 500 nm. The data acquisition parameters, such as laser power, gain in the amplifier and offset level, were fixed. All image analyses were performed using NIS-Elements software (Nikon, RRID:SCR_014329) and ImageJ2.

### 2.12. Quantification and Statistical Analysis

Cell counts were checked for normality using the Shapiro–Wilk statistical test. All values and graphs are expressed as mean ± SD. For comparison of 2 groups, unpaired two-tailed *t* tests were used. For comparison of 3 or more groups, one-way ANOVAs were preformed (GraphPad Prism). The threshold for statistically significant differences was set at a *p* value < 0.05.

## 3. Results

### 3.1. Fabrication and Characterization of PLCL Scaffold

The polymeric scaffolds were obtained by electrospinning PLCL copolymers with different compositions (70/30, 85/15 and 95/05 LA/CL) in the form of nanofibrous porous meshes. The solutions and the operating electrospinning conditions for each polymer were selected after optimization experiments aimed at obtaining bead-free fibers and are reported in the Section 2. Mats made of randomly oriented fibers were first produced by collecting the fibers on a drum rotating at low speed (see Section 2). SEM micrographs reported in Figure 1A revealed the presence of uniform and bead-free fibers for all the copolymers. The analysis of fiber diameter distribution showed that fibers with an average diameter of around 0.7 ± 0.2 μm were obtained for all samples (data not shown). The influence of the copolymer molar composition on thermal and mechanical properties of PLCL samples was investigated through calorimetric analysis and tensile testing, respectively. Figure 1B reports the overlay of the first DSC heating scans performed on the as-spun PLCL mats. The copolymers show a glass transition at temperatures (T_g_) of 22 °C, 42 °C and 57 °C for PLCL 70/30, PLCL 85/15 and PLCL 95/05, respectively, and a cold crystallization exothermic peak followed by an endothermic melting peak of the same entity for all samples. This result indicates that the melting phenomena that follow the cold crystallization concern only the PLCL crystal phase developed during the heating scan, demonstrating that completely amorphous PLCL mats were obtained during the electrospinning process, as previously reported for poly(L-lactic acid) [5]. As expected, the T_g_ values, as well as the melting temperature (T_m_) and melting enthalpy (ΔH_m_), of the crystal phase developed during the cold crystallization decreased with the increase in CL content in the copolymer. Stress–strain measurements of the PLCL scaffolds (Figure 1C) showed that an increase in the CL co-unit content from 5 mol% to 15 mol% caused a slight decrease in the elastic modulus from 89 MPa to 73 MPa, whereas the elastic modulus dropped down to 14.5 MPa when the content of CL units increased to 30 mol%. It is worth remarking that, among the studied samples, the PLCL 70/30 sample best approximates the mechanical behavior of SKM tissue (Young’s modulus of muscle fibers in relaxed state of 61 ± 5 kPa [36]) (Figure 1D). The results of the mechanical analysis, in particular the differences in the elastic modulus values of the samples, can be interpreted considering the solid-state properties of the copolymers [14]. Both PLCL 95/05 and PLCL 85/15 show a T_g_ higher than RT (57 °C and 42 °C, respectively); therefore, they are both glassy materials under the conditions of the mechanical measurements, showing a similar elastic modulus. PLCL 70/30, on the other hand, has a T_g_ lower than that of the other two copolymers (22 °C), in the RT range, which justifies its rubber-like behavior and corresponding lower stiffness. This latter copolymer was therefore selected to prepare scaffolds to be applied in SKM applications, given its mechanical properties. PLCL 70/30 scaffolds with aligned fibers were then fabricated using a drum rotating at high speed (see Section 2) to recapitulate the fibrous morphology of the SKM. SEM analysis revealed the presence of regular and bead-free fibers, while analysis with Fiji’s Directionality plugin showed fibers well aligned with the direction of rotation of the collector (Figure 1E, left image). A proportion of 68% of the total amount of nanofibers were aligned within a range of 0–18° from the direction of rotation, while only 2.6% were aligned within the range of 72–90°. Before biological studies, the scaffolds were sterilized with ethanol, then treated at 37 °C in buffer solution for 24 h (Figure 1E, middle image) to verify the dimensional and morphological stability of the scaffold during the cell culture tests and finally coated with type I collagen (Figure 1E, right image). SEM investigation confirmed that the fiber morphology was not altered either by the thermal treatment or by the collagen coating. Results of uncoated and collagen-coated PLCL are reported in Figure 1F, where the characteristic bands of PLCL and collagen functional groups can be identified; the absorption bands at 1750 cm^−1^ and 1730 cm^−1^ correspond to C-O in PLCL, and the absorption bands at 1650 cm^−1^ (amide I), 1560 cm^−1^ (amide II) and 3300 cm^−1^ (-OH) are attributed to the presence of collagen. WCA analysis (Figure 1G) confirmed the presence of collagen. Indeed, PLCL is a hydrophobic material characterized by a WCA value of 98.72 ± SD 3.1 (Figure 1G top image), whereas a value of 37.9 ± SD 3.5 was found for the collagen-coated electrospun mat (Figure 1G bottom image), indicating a significant increase in hydrophilicity.

### 3.2. PLCL Scaffold Influences Proliferation and Morphology of C2C12

To examine the cytocompatibility of the PLCL 70/30 scaffold, a cell viability test was assessed. C2C12 murine myoblasts were seeded and cultured on the collagen-coated PLCL scaffold prior to be fixed in a cell-crown insert (Figure 2A). Myoblast vitality and proliferation were analyzed at different time points (day 1, 5, and 7) and compared with the control. The analysis showed that viability was not affected by the presence of the PLCL scaffold over the culture period. A slight but not statistically significant difference in C2C12 proliferation rate was detected after 7 days, with lower cell growth on the PLCL scaffold compared to the control. In light of this result, we also analyzed cell morphology (Figure 2B). The morphology of C2C12 on the PLCL scaffold, as monitored by SEM, allowed us to study the different phases of cell adhesion, elongation and proliferation. We observed that in 2 h, C2C12 with round-shape morphology started to attach to the scaffold; in 6 h, C2C12 adhered completely and elongated following nanofiber orientation; and in 24 h, a layer of C2C12 covered the underlying nanofibrous surface (Figure 2C). These data are in accordance with previous studies [21,37], in which aligned fibers guided cell attachment and growth. Cell viability, proliferation and morphological analysis of C2C12 demonstrated the biocompatibility of the PLCL scaffold and proved the importance of unidirectional fiber orientation during C2C12 morphogenesis. 

### 3.3. Myogenesis of C2C12 Cultured on the PLCL Scaffold Differs from the C2C12 Control

With the notion that the nanostructure of our designed PLCL scaffold positively impacts the myoblast lineage, we monitored the expression of myogenic regulatory markers (MRFs) involved in myoblast fusion and myotube formation. The MRF transcript level was studied through the undifferentiated and differentiated phases of C2C12 on the PLCL scaffold and the C2C12 control. Pax7 is a key regulator of the embryonic skeletal myogenesis, and C2C12 on the PLCL scaffold and control stimulated to differentiate toward myogenic commitment (T1–T7) showed Pax7 downregulation; only day 7 displayed increased Pax7 expression on the PLCL scaffold due to the residual presence of undifferentiated cells (*p* < 0.05) (Figure 3A). Myf5 is expressed during the activation, proliferation and early differentiation stages of myogenesis, and in C2C12 on the PLCL scaffold is expressed since T1 showed a slightly higher level of Myf5 than the control (~2–2.5-fold) (*p* < 0.05), suggesting that the presence of the scaffold acted as further stimulus for cell activation and differentiation (Figure 3A). MyoD expression levels were constant over time under both conditions (Figure 3A). Together, Myf5 and MyoD activate myogenin (MyoG), which is associated with the final process of muscle differentiation and cell cycle arrest. We detected a significantly increased level of MyoG (*p* < 0.05) from almost threefold at T1 to fivefold at T7 in the PLCL scaffold compared to the control, affirming an early and accelerated differentiation (Figure 3B). At T1 Myf6 was overexpressed in the control relative to the PLCL scaffold, as an early expression occurred prior to muscle terminal differentiation and myogenesis. However, as expected, late Myf6 expression was higher at T3 for the PLCL scaffold as sign of anticipated differentiation (*p* < 0.05). Furthermore, our idea of early differentiation was highlighted by significantly higher expression of muscle ring-finger protein (MURF-1) in the PLCL scaffold than the control; MURF-1 is an autophagic marker present in the late stage of myogenesis when differentiated cells start to die (*p* < 0.05) (Figure 3B). We also analyzed desmin as a structural gene and first muscle-specific protein, whose expression levels remained rather constant over time under the two conditions, with a moderate induction in the control when cells were stimulated (*p* < 0.05) (Figure 3C). To complete our analysis, attention was focused on the myocyte-specific enhancer factor alpha factors (MEF2Cs) α1 and α2, which enhance or repress myoblast differentiation, respectively. In the PLCL scaffold, the ratio of Mef2Cα1/Mef2Cα2 suggested that Mef2Cα2 was predominant, thereby favoring myogenic commitment (*p* < 0.05) (Figure 3D). Overall, our data demonstrate that the presence of a PLCL scaffold significantly improves myogenesis, with significant changes in fundamental genes involved in the process.

### 3.4. Scaffold Accelerates Myocyte Fusion and Myotube Formation

To better understand the myogenesis of the cell-laden PLCL scaffold, confocal imaging of fluorescently stained myosin heavy chain 1 (MyHC-1) was assessed. MyHC-1 is a muscle-specific protein that started to increase or to be expressed from T5 to T10 [35]. In particular, at T0, undifferentiated C2C12 showed no cells positive for MyHc-1, whereas few differentiated cells positive for MyHC-1 were detected on the scaffold (Figure 4A). These data are consistent with our previous in vitro analysis, supporting the role of the PLCL scaffold in guiding C2C12 morphogenesis and accelerating cell organization with enhanced cell differentiation. At T1, more positive MyHC-1 cells were observed in C2C12 cultured on the PLCL scaffold (Figure 4B). At T3, as expected, fused myocytes and few myotubes were present (Figure 4C), whereas at T5, mature myotubes spread all over the scaffold (Figure 4D). In general, T5 in vitro is considered an intermediate time point of myogenesis; however, with the PLCL scaffold, it appeared as the final stage of differentiation. Stable myotubes were detected at T7, with a reduction in positive MyHC-1 confirming the hypothesis of an early differentiation pathway (Figure 4E). Moreover, Western blot (WB) analysis revealed that C2C12 on the scaffold exhibited a constant expression of MyHC-1, whereas the control exhibited MyHC-1 from T5 to T10 (Figure 4F). Collectively, a steady staining of MyHC-1 in aligned myotubes was observed for the PLCL scaffold throughout the cell lineage, proving that a scaffold can yield mature myoblasts within 5 days—half the time of the control.

### 3.5. Scaffold Induces Myoblast Interaction and Fusion with The Development of Cell–Cell Adhesions

Assuming that the PLCL scaffold influences C2C12 morphogenesis, morphological changes of C2C12 on the PLCL scaffold during myogenic commitment were also studied by SEM. The images taken at T0 showed the presence of myoblasts aligned according to scaffold orientation. Surprisingly, we noted the presence of membrane structures facilitating interactions and adhesions amongst myoblasts (Figure 5A). At T1, we witnessed an increased number of these membrane structures and assumed that the forced cell orientation driven by the topography of the scaffold coupled with the elasticity of the PLCL allowed for the establishment of cell–cell interactions via cell–cell adhesions and cell-to-cell connections. (Figure 5B). Consequently, at T5, fused myocytes were organized in mature myotubes marked by a progressive decline in cell–cell adhesion, (Figure 5C), and at T7, a larger myotube width and a significant reduction in membrane structures were observed (Figure 5D). The presented results revealed that growing C2C12 on a tissue-like matrix provides a more physiologically relevant construct that moves forward the cellular crosstalk with the activation of intracellular signaling pathways.

## 4. Discussion

SKM is a diffuse organ in our body, and the lack of its function or ability to regenerate leads to SKM disabilities and disorders [3]. SKM tissue consists of a collection of highly specialized cells grouped into muscular fibers made by the fusion of myocytes derived from precursor myoblasts. SKM development is regulated by complex signals, but the interaction between myoblasts, myocytes and myotubes during myogenesis is still under-researched [17]. Accordingly, muscle-like tissues obtained through TE represent a viable alternative to analyze myogenesis, overcoming the limitations of conventional in vitro and in vivo studies. Scaffolds play a key role in skeletal muscle tissue engineering (SMTE) and are designed to persist long enough to support functional SKM development [38,39]. To ensure good reproduction of SKM, we designed a scaffold made of biodegradable and biocompatible polymeric fibers characterized by a high degree of alignment obtained through the ESP technique [40,41,42]. To guarantee SKM functionality for our tailor-made electrospun scaffold, we identified, among a range of biopolymers, the aliphatic polyester PLCL. Specifically, taking into account the mechanical properties of different PLCL copolymers, we chose PLCL 70/30, with a higher content of CL co-units and with an elastic modulus of about 14.5 MPa, which is optimal for muscle fibers, to mimic the native tissue microenvironment crucial for myogenic differentiation [43]. Next, the PLCL scaffold was coated with collagen to improve cell adhesion through surface recognition, reproducing a tissue-like environment [28]. Once the nanostructure and the chemical composition of the scaffold were selected, we unraveled which signaling regulated SKM development, influencing myoblast morphology and activity [44,45].

It is crucial to study myogenesis with a myoblast cell line in order to clearly monitor the stages of translation from a single myoblast into an elongated myocyte up to a fused myotube, also guaranteeing the formation of an SKM analogous to that developed in vivo. In this study, we use the C2C12 murine myoblast cell line and first examined the cytocompatibility of the designed scaffold, a key step in the development of the biomaterial. Cell viability showed that C2C12 myoblasts cultured on the PLCL scaffold had a similar proliferation rate as the C2C12 control cultured on the TCP. By performing SEM analysis, we found that the topography of the scaffold guides C2C12 morphogenesis. Interestingly, within 2 h, undifferentiated cells were already attached to the scaffold; within 6 h, they were elongated, and within 24 h, they were able to cover the nanofibrous substrate with a confluent cell layer. In agreement with previous studies [21,37], we confirmed that the contact guidance of the scaffold provided linear attachment points for cell adhesion, orientation and elongation and, moreover, facilitated the organization of C2C12. This method allowed us to easily analyze the key stages prior to tissue formation; we could observed at which point within 24 h the immature myoblast, when placed in contact with the substrate recreating a SKM tissue-like structure, began to interact with the external environment, in the absence of exogenous induction [24]. Moreover, it is important to demonstrate how myoblast differentiation is driven because it is possible to detect which factors contribute significantly towards this process and are central to an understanding of how SKM develops in different environments or under pathological circumstances. Consequently, we analyzed the whole process, and by looking at the genetic profile of the cells throughout the differentiation pathway, we found that an early regulatory gene of myogenesis, Myf5, was more present in C2C12 on the scaffold than in the control. Myf5 works with MyoD, which initiates myoblast determination and regulates downstream genes such as MyoG, and was upregulated in C2C12 on the scaffold. The increased expression level of Myf5 and, consequently, that of MyoG suggested that the PLCL scaffold acts in this first phase of myogenic commitment. Subsequently, MyoD and MyoG shut down the expression of Myf5 and stimulate the expression of Myf6, which participates belatedly when the SKM is mature; it was therefore unsurprising that late Myf6 expression occurred earlier on the scaffold. Below, we evaluate the expression of the myocyte-specific enhancer factor alpha factors (MEF2C), studying the ratio between Mef2cα1, which recruits HDAC5 to repress muscle-specific genes, and Mef2Cα2, which promotes myoblast differentiation [46,47]. The Mef2Cα1/Mef2Cα2 ratio in C2C12 on the scaffold indicated a predominant Mef2Cα2 value, which promotes myoblast maturation. Analyses of the structural gene desmin, a key intermediate filament subunit in the SKM sarcomere that is critical in maintaining the mechanical integrity of the cell during contraction, displayed a comparable trend under both cell culture conditions. This result confirmed that the scaffold modulates differentiation without changing the structure of vital protein. We also investigated the expression of MURF-1, an autophagic marker present in the advanced state, at which point differentiated cells begin to die [48], and MURF-1 was found to be overexpressed in the PLCL scaffold compared with the control. These analyses supported our theory that the scaffold positively influences the behavior of myoblasts by stimulating the early stage of differentiation. This discovery is a great asset, as we obtained a mature SKM in a reduced amount of time. These results also make clear the fact that the 2D cell culture approach is not optimal for studying tissue formation, as it is poorly representative of in vivo conditions. In accordance with the genetic profiles, the protein expression of MyHC-1 by IF and WB was also observed earlier during the process, demonstrating a more pronounced and anticipated cell commitment with adequate myoblast-to-myotube formation within a shorter time frame On the whole, the PLCL scaffold facilitated C1C12 myogenesis with high and early expression levels of myogenic regulatory factors and downstream propagation signaling, as shown by the patterning of desmin and MyHC-I. Furthermore, we studied this phenomenon by performing SEM analysis of undifferentiated and differentiated C2C12 and turned our attention to the interactions of myoblasts by detecting the presence of microvilli-like structures that originate from the cell membrane and act as bridges between cells [49,50,51,52]. Curiously, an increased presence of such membrane structures was observed as early as after 1 day of differentiation; myoblasts were attached and elongated on the scaffold. At T5, we observed fused myocytes already organized to form mature myotubes and a significant reduction in such membrane structures, and at T7, mature myotubes with no membrane structures were observed. Together with our previous analyses, such findings illustrate the role of the material cues, from both biophysical and biochemical perspectives, in boosting the development of SKM tissue in a short time frame, as the scaffold promotes the formation of cell adhesions, which are important during cell differentiation. We demonstrated how cells, in the early stage of myogenesis and grown in the appropriate microenvironment, differentiate, forming cell–cell bonds to release molecules capable of triggering the inner signaling pathway implicated in the activation of SKM maturation. Consequently, when myogenic maturation and myogenic differentiation are completed, the microvilli-like membrane structures gradually disappear. Indeed, these results demonstrate what, according to the literature, is not a well-known mechanism [53,54]. In recent years, researchers have reported that cell adhesions, through direct and indirect transfer of factors, can regulate myogenesis through activation of signaling and secretion of cytokines, which then promote cell differentiation and protein synthesis [55]. Here, we showed that cells were dependent on membrane structures that are important in the regulation of progression from myoblasts to myocytes into mature myotubes; more research will be performed to detect the cell–cell adhesions that are formed in this phase, as well as to recognize the effects of the released molecules.

Overall, in the current study, we elucidated the benefits of growing C2C12 on an electrospun scaffold coated with collagen from both morphological and molecular perspectives, suggesting how our PLCL scaffold provides a more suitable microenvironment for the fabrication of physiological SKM in vitro. The detailed data obtained from gene and protein analyses, in accordance with results of SEM, indicate how myoblast fusion is synchronized with the development of cell–cell adhesions, which can activate and anticipate multiple signaling pathways essential during myogenesis [56]. Although TE has been criticized for the simplicity of the system, which is unable to fully reproduce the native complexity of human tissues, recent studies have shown the potential of producing implantable biomedical scaffolds with hierarchical morphology of tissues [57] to disrupt the way regenerative medicine is considered [58]. Here, we have highlighted how a scaffold adapted to simulate the desired tissue microenvironment represents an alternative and efficient tool to study myogenic differentiation, with a significant improvement in the differentiation and maturation process, recreating a physiological SKM tissue in vitro. Further studies will investigate the role of mechanical transduction of the scaffold during SKM formation to understand which signaling pathways have contributed to changes in C2C12 morphology and myogenic differentiation. Additionally, in vivo studies will fabricate vital SKM tissue, reproducing the neurovascular compartment of the neuromuscular unit. Although TE is still subject to important challenges concerning its in vivo applications, it can be considered a novel strategy for in vitro tissue formation and functional testing—processes that required to achieve success in vivo and in the clinic [20]. Such an in vitro engineering method, which is easy and cheap to reproduce, enables the detailed study of tissue development by revealing the various steps of myogenesis in a short period of time. The versatility and reproducibility of the system could be exploited to fabricate in vitro tissue models under diseased conditions in order to investigate pathogenetic mechanisms and test the activities of new drugs, thereby speeding up preclinical testing with a reduced time cost.

## Figures and Tables

**Figure 1 bioengineering-10-00239-f001:**
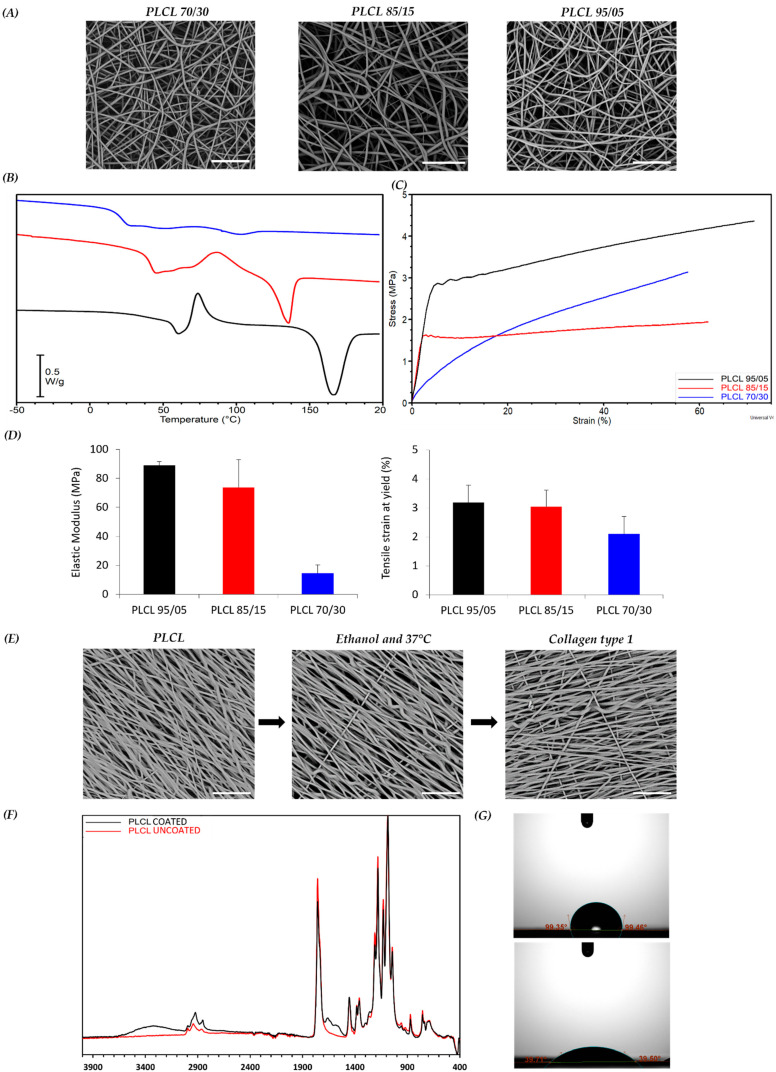
Scaffold characterization. (**A**) SEM micrographs of scaffolds: PLCL 70/30 (blue), 85/15 (red) and 95/05 (black). (**B**) DSC heating ramp of scaffolds and glass transition temperature (T_g_). (**C**) Representative stress-strain curves of scaffolds. (**D**) Young’s modulus and yielding points of scaffolds. (**E**) SEM micrographs of PLCL 70/30 scaffold after sterilization with ethanol, treatment at 37 °C and coating with type I collagen. (**F**) ATR-IR spectra of coated and uncoated PLCL mats. (**G**) WCA measurements for uncoated (**top**) and coated (**bottom**) PLCL mats. Scale bars: 20 µm.

**Figure 2 bioengineering-10-00239-f002:**
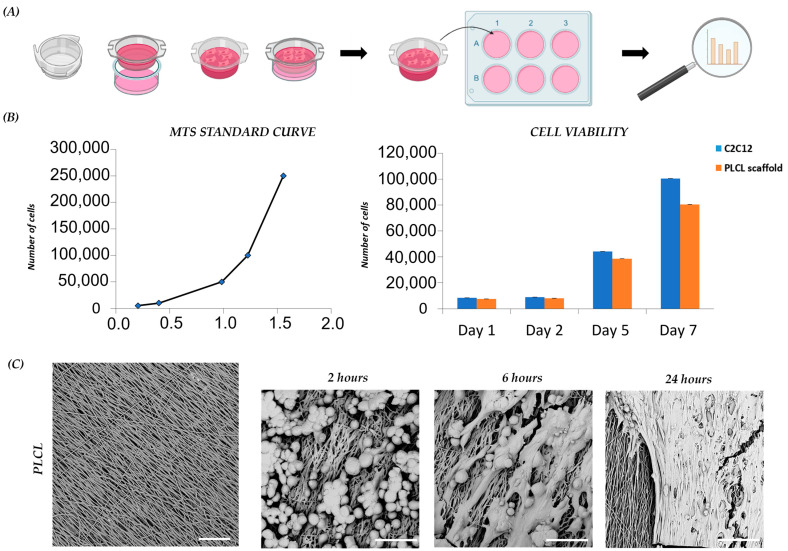
C2C12 proliferation and morphology on scaffolds: (**A**) schematic diagram of PLCL scaffold preparation for C2C12 culture; (**B**) C2C12 viability and proliferation on scaffold and control on day 1, day 2, day 5 and day 7 of culture; (**C**) representative SEM images of PLCL scaffold before and after cell seeding, capturing C2C12 morphology after 2, 6 and 24 h. Scale bars: 25 µm.

**Figure 3 bioengineering-10-00239-f003:**
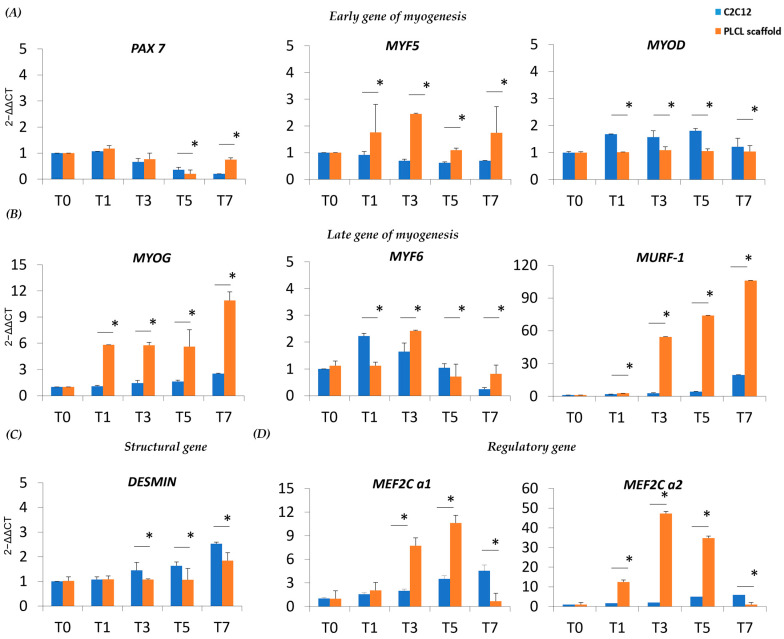
Real-time q-PCR of MRFs, muscle-specific proteins and MEF transcript levels in C2C12 on the scaffold and C2C12 control during myogenic differentiation. (**A**) Transcript levels of early MRFs and (**B**) late MRFs immediately after 24 h of differentiation (T1), during differentiation (T3–T5) and during late stages of differentiation (T7). (**C**) Expression of a structural gene after 24 h of differentiation (T1), during differentiation (T3–T5) and during late stages of differentiation (T7). (**D**) Expression of MEF after 24 h of differentiation (T1), during differentiation (T3–T5) and during late stages of differentiation (T7). * Data are representative of three experiments and expressed as means ± SD; the level of significance was set at *p* < 0.05.

**Figure 4 bioengineering-10-00239-f004:**
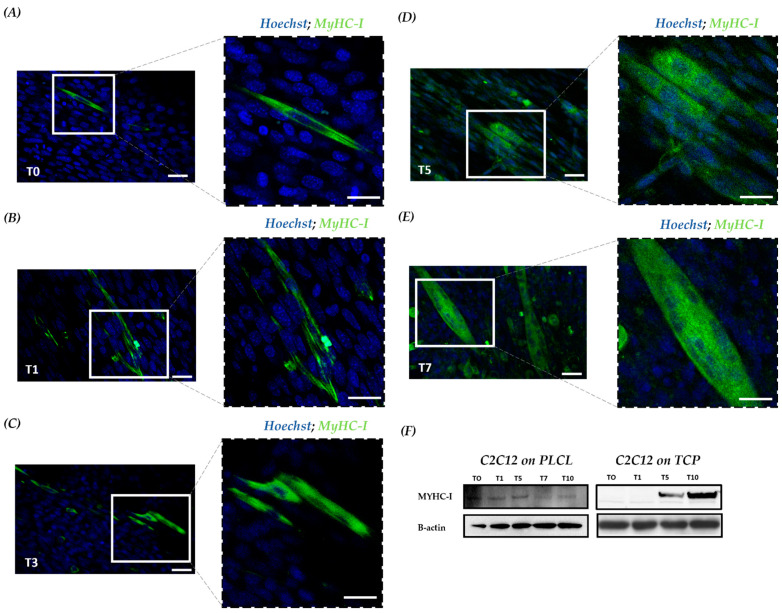
MyHC-1 expression during myogenic commitment of C2C12 on a scaffold. Confocal microscopy investigation with IF staining of the cell-based scaffold; MyHC-1 is indicated in green, and nuclei are counterstained with DAPI in blue. (**A**) At T0, undifferentiated cells only stained for nuclei in blue and differentiated with a cytoplasmic green signal positive for MyHC-1. Scale bars: 25 μm and 10 μm. (**B**) At T1, a progressive increase in differentiated cells positive for MyHC-1. (**C**) At T3, undifferentiated cells and fused myocytes with initial myotube formation. (**D**)At T5, increasing number of myotube formation (**E**) At T7, mature multinucleated myotubes. (**F**) WB for MyHC-1 in total protein fractions; the bands were quantitated by calculating the relative quantities of MyHC-1 normalized to actin. Data are representative of three experiments and expressed as means.

**Figure 5 bioengineering-10-00239-f005:**
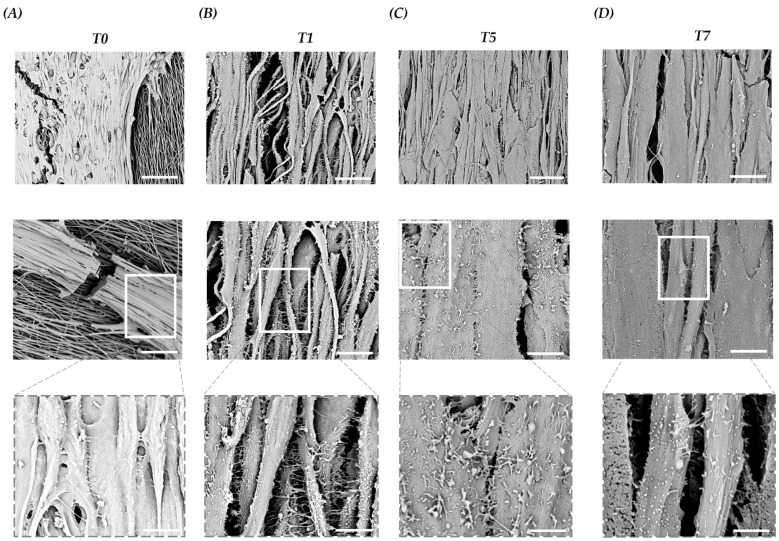
Representative SEM images of C2C12 on a scaffold at T0 and during differentiation (T1–T7). (**A**) At T0, aligned C2C12 myoblasts focus on cell–cell interactions and membrane structures. Scale bars: 50 µm, 15 µm and 5 µm. (**B**) At T1, C2C12 myocyte interactions and development of cell–cell adhesions with increased membrane structures. Scale bars: 15 µm, 5 µm and 2.5 µm. (**C**) At T5, myocyte fusion and myotube formation with lower membrane structures. Scale bars: 15 µm, 5 µm and 1.5 µm. (**D**) At T7, enlarged size of myotubes with reduced membrane structures. Scale bars: 10 µm, 5 µm and 1.5 µm.

**Table 1 bioengineering-10-00239-t001:** Forward (Fw) and reverse (Rev) primers for real-time qPCR analysis.

Gene (GenBank Accession Number) (Mus Musculus)	Primers
Myf5 Myogenic Factor 5 (NM_008656.5)	Fw: 5’-AGGTGGAGAACTATTACAGC
	Rev: 5’-TGATACATCAGGACAGTAGATG
Myog Myogenin (NM_031189.2)	Fw: 5’-AGTACATTGAGCGCCTAC
	Rev: 5’-CAAATGATCTCCTGGGTTG
Des Desmin (NM_010043.2)	Fw: 5’-ACACCTAAAGGATGAGATGG
	Rev: 5’-GAGAAGGTCTGGATAGGAAG
Murf-1(TRIM63) Muscle-specific RING finger protein 1 (NM_001039048.2)	Fw: 5’-GACTTAGAACACATAGCAGAG
	Rev: 5’-CTCTTCTGTAAACTCCTCCTC
Myf6 Myogenic Factor 6 (NM_001003982.1)	Fw: 5’-ATAGAGAAGGAGCCGTGTTGG
	Rev: 5’-TTCTCTGAGATCTGGCTGGGA
Mef2c isoforma α1 Myocyte Enhancer Factor 2 C (NM_001170537.1)	Fw: 5’-CTCAGACATTGTGGAGACATT
	Rev: 5’-TCAGGGCTGTGACCTACTG
Mef2c isoforma α2 Myocyte Enhancer Factor 2 C (NM_001347568.1)	Fw: 5’-CTCAGACATTGTGGAGGCAT
	Rev: 5’-TTCTTCAGTGCGTGGGGT
GAPDH Glyceraldehyde-3-Phosphate Dehydrogenase (NM_001256799.2)	Fw: 5’-CTCTGATTTGGTCGTATTGG
	Rev: 5’-GTAAACCATGTAGTTGAGGTC

## Data Availability

Not applicable.

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
