# Peer review of "Electrospun Poly(L-lactide-co-ε-caprolactone) Scaffold Potentiates C2C12 Myoblast Bioactivity and Acts as a Stimulus for Cell Commitment in Skeletal Muscle Myogenesis"

_bioengineering, 2023, doi:10.3390/bioengineering10020239_

Round 1
Reviewer 1 Report
The article entitled “Synthetic electrospun poly(L-lactide-co- ε-caprolactone) scaffold for skeletal muscle myogenesis: molecular and morphological analysis on C2C12 cell line" by Pacilio Serafina. et.al addresses current problem of skeletal muscle regeneration, focusing on effects of the polymer based-scaffold on the differentiation pathway at different stages of myogenesis using accurate quantitative analysis of differentiation-specific markers. The general characterization of obtained scaffolds are correctly presented and advanced and deep cell studies proving the concept of the work. Nevertheless, I ask the authors to respond to the following comments.
Major comments:
1. Modification of PLCL surface with collagen type I seems to be standard procedure in the presented protocol. In other studies related to polyesters surface modification activation of the surface e.g. by plasma treatment is applied in order to increase the interactions between the collagen and polymer functional groups. In presented work the coating seems to be only the physical type of modification. Do the authors have results confirming the presence of collagen on the surface of the fibers?
Minor comments:
1. The order in Materials and methods section suggest that sterilization was performed on coated samples, whereas from the text shows that samples were coated after sterilization. Please make in the right order.
2. Please uniform the style of numbers e.g. line 111, 241 should be “.” instead of “,” . The same on the scales Fig. 1 and 5.
3. In section 2.4 TGA is described as one of the methods for scaffold characterization but the article does not contain results from this method.
4. Wrong text formatting in paragraph 3.5 (italic).
5. In some areas of text “in vitro” it is italicized in others it is not, please standardize.
Author Response
Bologna, 17/01/2023
Manuscript ID number: bioengineering-2148991
Type of manuscript: Article
Title: Synthetic electrospun poly(L-lactide-co- ε-caprolactone) scaffold for
skeletal muscle myogenesis: molecular and morphological analysis on C2C12
cell line
Dear Editor,
we would like to thank you for offering us the opportunity to resubmit our manuscript following a revision of the text. We would also thank the reviewers for their comments, thanks to which the impact of our findings may be better elucidated and discussed.
Herein we report the revised manuscript, following the guidance of reviewer comments. Every change made in the manuscript has been highlighted in yellow, we numbered the lines in the manuscript in order to facilitate reviewers’ correction.
We hope this revised version of the manuscript is acceptable to the Referees and the Editors, and that it is now suitable for publication.
Please find below a point-by-point review of our response with regards to the question raised by the reviewers.
Reviewer #1: The article entitled “Synthetic electrospun poly(L-lactide-co- ε-caprolactone) scaffold for skeletal muscle myogenesis: molecular and morphological analysis on C2C12 cell line" by Pacilio Serafina. et.al addresses current problem of skeletal muscle regeneration, focusing on effects of the polymer based-scaffold on the differentiation pathway at different stages of myogenesis using accurate quantitative analysis of differentiation-specific markers. The general characterization of obtained scaffolds are correctly presented and advanced and deep cell studies proving the concept of the work. Nevertheless, I ask the authors to respond to the following comments.
Concerns:
1. Modification of PLCL surface with collagen type I seems to be standard procedure in the presented protocol. In other studies related to polyesters surface modification activation of the surface e.g. by plasma treatment is applied in order to increase the interactions between the collagen and polymer functional groups. In presented work the coating seems to be only the physical type of modification. Do the authors have results confirming the presence of collagen on the surface of the fibers?
We thank the referee for this comment, we have added results confirming the presence of collagen on the surface of the fibers. PLCL, as described in the Materials and Methods section in paragraphs 2.4 and 2.5, after the sterilization process, was coated with collagen type one prior to C2C12 seeding. We perform two analyses to demonstrate the presence of collagen coating: Fourier Transform Infrared spectroscopy – Attenuated Total Reflectance (FTIR-ATR) and Water Contact Angle (WCA).
We added in paragraph 2.5. in the Materials and Methods section, the experimental part related to FTIR-ATR and WCA techniques, as follows:
“FT-IR was carried out by Spectrum Two instrument equipped with ATR accessory (Perkin-Elmer, diamond crystal, Milan, Italy) on both uncoated and coated scaffolds. All spectra have been registered between 400 cm−1 and 4000 cm−1 with a resolution of 4 cm−1, accumulation 16 scans and step size 1 cm−1. Water contact angle (WCA) was assessed by the sessile drop method in air, time of analysis of 10 seconds, using a KSV CAM101 instrument (KSV Instruments Ltd., Helsinki, Finland) on uncoated and coated scaffolds. Ten measurements were performed for each sample.” (lines 169-175)
We added in paragraph 3.1 and in the corresponding Figure 1. (F and G) in the Results section the results confirming the presence of collagen, as follows:
“Results of uncoated and collagen-coated PLCL are reported in Figure 1F, where the characteristic bands of PLCL and collagen functional groups can be identified: the absorption bands at 1750 cm−1 and 1730 cm−1 corresponding to C–O in PLCL, the absorption bands at 1650 cm−1(amide I), 1560 cm−1 (amide II) and 3300 cm-1(-OH) attributed to the presence of collagen. WCA analysis (Figure 1G) confirmed the presence of collagen. Indeed, PLCL is a hydrophobic material characterized by a value of WCA of 98.72 ± SD 3.1 (Figure 1G top image), whereas a value of 37.9 ± SD 3.5 was found for the collagen-coated electrospun mat (Figure 1G bottom image), indicating a significative increase of hydrophilicity.” (lines 292-300)
- The order in Materials and methodssection suggest that sterilization was performed on coated samples, whereas from the text shows that samples were coated after sterilization. Please make in the right order.
We thank the referee for the advice, and we modified the paragraph order in the Material and Methods section as follows:
“2.3. Scaffold Characterization Methods
2.4. Cell Seeding Conditions
2.5. Scaffold Functionalization”.
(line 144; 161; 165)
Adjusted order paragraph from 2.9. , 2.10. , 2.11. , 2.12.
Please uniform the style of numbers e.g. line 111, 241 should be “.” instead of “,” . The same on the scales Fig. 1 and 5.
We thank the referee for the advice; we modified the style number:
Line (125,135,199,201,278,426,427,429,)
The same on the graph scale of Figure (Figure 1D, Figure 2B, Figure 3). Figure 1,2 and 3 were replaced.
In section 2.4TGA is described as one of the methods for scaffold characterization but the article does not contain results from this method.
We thank the referee for the observation. When we reread the article we found that TGA results were not relevant, and we decided to delete them in the Results section. According to this, we deleted TGA in the Material and Methods section, paragraph 2.3. (lines 150-152)
Wrong text formatting in paragraph 3.5 (italic).
We thank the referee; we corrected the text format of paragraph 3.5.
- In some areas of text, “in vitro” it is italicized in others, it is not, please standardize.
We thank the referee; we corrected the word style (lines 47, 48, 50).
Sincerely yours,
Prof.ssa Giovanna Cenacchi, MD
Lab Patologia e Diagnostica Subcellulare, DIBINEM

Reviewer 2 Report
The work entitled “Synthetic electrospun poly(L-lactide-co- ε-caprolactone) scaffold for skeletal muscle myogenesis: molecular and morphological analysis on C2C12 cell line” provides a lot of information about the effects of substrate on myogenic behaviours of C2C12. Overall, the work is scientifically sound. However, I have some suggestions before the manuscript has considered to be publish.
Several literatures have been investigated the effect of poly(L-lactide-co-ε-caprolactone) on the regulation of cell myogenesis, such as Biomaterials. 2004, 25(28):5939-46, J Biomed Nanotechnol. 2017, 13(3):303-12, Acta Biomater. 2012, 8(2):531-9, J Biomater Sci Polym Ed. 2012. In addition, there are many studies related to electrospun poly(L-lactide-co- ε-caprolactone) scaffold. Therefore, the method of preparation scaffolds and the effects of materials on cellular response present in this work do not show significant different from previous studies. The authors have to emphasize the novelty of this work and compare their results with previous works.
Author Response
Bologna, 17/01/2023
Manuscript ID number: bioengineering-2148991
Type of manuscript: Article
Title: Synthetic electrospun poly(L-lactide-co- ε-caprolactone) scaffold for
skeletal muscle myogenesis: molecular and morphological analysis on C2C12
cell line
Dear Editor,
we would like to thank you for offering us the opportunity to resubmit our manuscript following a revision of the text. We would also thank the reviewers for their comments, thanks to which the impact of our findings may be better elucidated and discussed.
Herein we report the revised manuscript, following the guidance of reviewer comments. Every change made in the manuscript has been highlighted in yellow, we numbered the lines in the manuscript in order to facilitate reviewers’ correction.
We hope this revised version of the manuscript is acceptable to the Referees and the Editors, and that it is now suitable for publication.
Please find below a point-by-point review of our response with regards to the question raised by the reviewers.
Reviewer #2: The work entitled “Synthetic electrospun poly(L-lactide-co- ε-caprolactone) scaffold for skeletal muscle myogenesis: molecular and morphological analysis on C2C12 cell line” provides a lot of information about the effects of substrate on myogenic behaviours of C2C12. Overall, the work is scientifically sound. However, I have some suggestions before the manuscript has considered to be publish.
Several literatures have been investigated the effect of poly(L-lactide-co-ε-caprolactone) on the regulation of cell myogenesis, such as Biomaterials. 2004, 25(28):5939-46, J Biomed Nanotechnol. 2017, 13(3):303-12, Acta Biomater. 2012, 8(2):531-9, J Biomater Sci Polym Ed. 2012. In addition, there are many studies related to electrospun poly(L-lactide-co- ε-caprolactone) scaffold. Therefore, the method of preparation scaffolds and the effects of materials on cellular response present in this work do not show significant different from previous studies. The authors have to emphasize the novelty of this work and compare their results with previous works.
We thank the referee for the comment; we added more details on the present studies regarding the use of electrospun poly(L-lactide-co- ε-caprolactone) scaffolds in tissue engineering applications. In order to make the present study clearer and more affordable, we added some relevant information comparing our work with previous studies in the Introduction section in paragraph 1, as follows:
“Since in their native environment cells are surrounded by different structures with different scales, it is not surprising that cell attachment, proliferation and differentiation could be affected due to presence of external micro-patterned material [24]. However, external stimuli must be constant in order to have an impact on cells; consequently, a scaffold must maintain its mechanical integrity. It is well-known that copolymerization of lactide (LA) with the flexible caprolactone (CL) unit is an appropriate method for controlling mechanical properties and degradation rate of the resulting materials, [25], [26] widening the range of biomedical applications of this family of polymers, and making them ideal materials for conducting steady stimuli in vitro or in vivo [27]. Accordingly, in the present work, we have designed an electrospun scaffold made of poly(L-lactide-co-ε-caprolactone) (PLCL) further coated with type I collagen to recreate a favourable environment for cells which, in recognising the collagen as the main constituent of the ECM of SKM, succeed in adhering to the substrate [28]. The scaffold is characterized by controlled microarchitecture to promote cell growth, by suitable physiochemical properties to favour cell interactions, by the right mechanical properties to guarantee cell functionality, and by a customized morphology to resemble the native environment, all of which, taken together, allowed us to further study the phenotypic profile of myoblasts throughout SKM tissue formation. As suggested by E. Martinez [29] and Huaqiong Li [24] micro- and nanostructure can lead to deformation of cells and, reflexively, the nucleus, which is attached by intermediate filaments to the cytoskeleton. Thus, we explored the effects of the polymer based-scaffold on the differentiation pathway at different stages of myogenesis using accurate quantitative analysis of differentiation-specific markers, examining Myogenic Regulatory Factors (MRFs) that directly control the myogenic process and interact with Myocyte-specific enhancer factor (MEFs), which regulate myoblast commitment [30], [31]. Furthermore, myoblast morphology towards myogenic commitment was studied under both confocal and electron microscopy to figure out the role of the scaffold in shaping SKM myogenesis and tissue growth, by detecting the interactions that modulate myoblasts, myocytes and myotubes formation and so, unravelling the link between cell fate and material cue.” (lines 78-113)
Additionally, we clarified and remarked the novelty of our data at the Discussion section in paragraph 4, as follows:
“It is crucial to study myogenesis with a myoblast cell line in order to clearly monitor the stages of translation from a single myoblast into an elongated myocyte up to a fused myotube, guaranteeing also the formation of a SKM analogous to that developed in vivo. In this study, we use the C2C12, murine myoblast cell line and we first examined the cytocompatibility of the designed scaffold, a key step in the development of the biomaterial.” (lines 452-455)
“This method allowed us to easily analyse the key stages prior to tissue formation, where the immature myoblast, when placed in contact with the substrate, which recreates a tissue-like structure, begins to interface with the external environment forming in 24 hours, in absence of exogenous induction, the SKM tissue micro-architecture [24]. Moreover, it is important to demonstrate how myoblast differentiation is driven, because it is possible to detect which factors contribute significantly towards this process and are central to an understanding of how SKM develops under different environments or under pathological circumstances” (lines 465-472)
“This discovery is a great asset as we obtained mature SKM in a reduced amount of time; it has also made clear that the 2D cell culture approach is not optimal for studying tissue formation, as it is poorly representative of the in vivo conditions. In accordance with the genetic profiles, the protein expression of MyHC-1 by IF and WB, was also observed earlier during the process, demonstrating a more pronounced and anticipated cell commitment with adequate myoblast-to-myotube formation within a shorter time frame” (lines 494-500)
“Here we showed that cells were dependent on membrane structure, that are important in the regulation of progression from single myoblast to myocyte into mature myotubes; more research will be performed to detect the cell-cell adhesions that are formed in this phase as well as to recognize the effects of the released molecules” (lines 523-527)
“…suggesting how our PLCL scaffold provides a more suitable microenvironment for the fabrication of physiological SKM in vitro. The detailed data obtained from gene and protein analyses, in accordance with results performed by SEM, figured out how myoblast fusion is synchronized with the development of cell-cell adhesions that could activate and anticipate multiple signalling pathways essential during myogenesis” (lines 531-535)
To exhaustively answer we added the following references:
[24] H. Li et al., ‘Direct laser machining-induced topographic pattern promotes up-regulation of myogenic markers in human mesenchymal stem cells’, Acta Biomaterialia, vol. 8, no. 2, pp. 531–539, Feb. 2012, doi: 10.1016/j.actbio.2011.09.029.
[25] H. Qian, J. Bei, and S. Wang, ‘Synthesis, characterization and degradation of ABA block copolymer of l-lactide and ε-caprolactone’, Polymer Degradation and Stability, vol. 68, no. 3, pp. 423–429, May 2000, doi: 10.1016/S0141-3910(00)00031-8.
[26] J. P. Penning, H. Dijkstra, and A. J. Pennings, ‘Preparation and properties of absorbable fibres from l-lactide copolymers’, Polymer, vol. 34, no. 5, pp. 942–951, Mar. 1993, doi: 10.1016/0032-3861(93)90212-S.
[27] S. I. Jeong et al., ‘In vivo biocompatibilty and degradation behavior of elastic poly(L-lactide-co-epsilon-caprolactone) scaffolds’, Biomaterials, vol. 25, no. 28, pp. 5939–5946, Dec. 2004, doi: 10.1016/j.biomaterials.2004.01.057.
[28] T. Wu et al., ‘Development of Dynamic Liquid and Conjugated Electrospun Poly(L-lactide-co-caprolactone)/Collagen Nanoyarns for Regulating Vascular Smooth Muscle Cells Growth’, J Biomed Nanotechnol, vol. 13, no. 3, pp. 303–312, Mar. 2017, doi: 10.1166/jbn.2017.2352.
[29] E. Martínez et al., ‘Stem cell differentiation by functionalized micro- and nanostructured surfaces’, Nanomedicine (Lond), vol. 4, no. 1, pp. 65–82, Jan. 2009, doi: 10.2217/17435889.4.1.65.
Sincerely yours,
Prof.ssa Giovanna Cenacchi, MD
Lab Patologia e Diagnostica Subcellulare, DIBINEM

Reviewer 3 Report
This work present a molecular and morphological analysis of synthetuc electrospun scaffolds that are interesting right now in the engineering sector (specifically in tissue engineering). In this way, this work is novel, interesting and suitable for its publication in this journal. Nevertheless, some comments must be solved before its publication:
-Section 2.1: Is it possible to supply the molecular weight of the copolymers? This weight is decisive in its electrospinning process.
-Line 106-114: Why those electrospinning parameters were selected? An explanation of their selection or a bibliographic reference must be included.
-How scaffold functionalization was made? More information about this must be included in the manuscript. Why is this better than incorporate the collagen directly in the electrospun solution? (https://doi.org/10.1002/jbm.a.37156)
-Line 134: Mechanical tests were performed in the nanofibers directions. What about the perpendicular direction? This direction will also support stresses and its resistance will be much less. It is important to consider it also in these tests (https://doi.org/10.1007/s10704-020-00460-4).
-Figure 1C: It is also interesting to discuss about the difference in the tenacity of each scaffold.
-Line 241: I don't agree with this sentence. PLCL 70/30 seems to have a change in slope (from elastic to plastic) much sooner (about 2%). Please, verify this.
-Lines 242-246: This sentence need more explanation and a reference that support it.
-Line 250: The alignament of the sample could be measured by FIJI.
-Lines 253-254: Has it been confirmed that this immersion impregnates the sample with Collagen? Where is the collagen: integrated into the fibers or in the holes between them? Has it been checked for interactions between the copolymers and collagen? Is the collagen distributed evenly throughout the sample? Does the sample dry between this immersion and the deposition of the cells? I believe that this method for incorporating collagen into the samples is interesting, but there are still many doubts to be resolved in it that can condition the biological behavior of the samples. If these questions have been solved in other work. This must be referenced.
-Figure 1E: This process is misunderstanding in the methodology section. Were samples firstly sterilized by ethanol and later inmersed in collagen solutions?
Author Response
Bologna, 31/01/2023
Manuscript ID number: bioengineering-2148991
Type of manuscript: Article
Title: Synthetic electrospun poly(L-lactide-co- ε-caprolactone) scaffold for skeletal muscle myogenesis: molecular and morphological analysis on C2C12 cell line
Dear Reviewer,
thank for your comments. Please find below a point-by-point review of our response with regards to the question raised.
Reviewer #3: This work present a molecular and morphological analysis of synthetuc electrospun scaffolds that are interesting right now in the engineering sector (specifically in tissue engineering). In this way, this work is novel, interesting and suitable for its publication in this journal. Nevertheless, some comments must be solved before its publication.
Concerns:
1. Section 2.1: Is it possible to supply the molecular weight of the copolymers? This weight is decisive in its electrospinning process
In Section 2.1, the molecular weight of the copolymer is indicated in terms of inherent viscosity. It is well accepted, in polymer science, that inherent (intrinsic) viscosity is a good practical measure of the weight-average molecular weight of a polymer. Indeed, the value of the inherent viscosity is the only indication of the molecular weight, provided by the Corbion company where the copolymers were purchased.
Line 126-133: Why those electrospinning parameters were selected? An explanation of their selection or a bibliographic reference must be included.
We thank the referee for the valuable comment. We agree that an explanation of the electrospinning parameters selection is necessary. We included the sentence below, in the Results section, at the beginning of paragraph 3.1. Fabrication and Characterisation of PLCL Scaffold:
“The electrospun solutions and the operating conditions for each polymer were selected after optimization experiments aimed at obtaining bead-free fibers and are reported in the Materials and Methods section.” (lines 260-262)
Why the used funtionalization method is better than incorporate the collagen directly in the electrospun solution? (https://doi.org/10.1002/jbm.a.37156)
We decided to use the collagen surface coating procedure, instead of using a blend of collagen and the synthetic polymer, because our aim was to modify only the surface of the fibres, without affecting the bulk properties, and in particular, the mechanical properties of the fibres. Indeed, our goal was to improve cell adhesion through surface recognition.
In previous works, we prepared blends of polylactic acid and collagen (incorporating the collagen in the electrospun solution directly), but our aim was different with respect to that of the present work, since we aimed at fabricating hybrid fibres made of natural and synthetic polymer in the bulk. In this case we observed mechanical properties that varied with fibre composition (Biofabrication 9 (2017) 015025 https://doi.org/10.1088/1758-5090/aa6204).
Line 146: Mechanical tests were performed in the nanofibers directions. What about the perpendicular direction? This direction will also support stresses and its resistance will be much less. It is important to consider it also in these tests (https://doi.org/10.1007/s10704-020-00460-4).
We thank the referee for the comment. We used fibers oriented in a single direction in order to preserve the skeletal muscle's (SKM) biological structure. In this case, a perpendicular direction would have impaired the myoblast structure and, consequently, the SKM development. Uniaxial direction alone is vital for determining the myoblast organization underlying SKM development, as shown in our manuscript. Furthermore, considering the hierarchical structure of the SKM, in which each myofibril contains contractile proteins arranged longitudinally in sarcomeres, the uniaxial direction of the nanofibers allows the isometric contraction with a structure maintenance as it is observed in the in vivo condition; since myosin, which constitutes the thick myofilaments, binds and slides actin, the thin myofilaments, maintaining the longitudinal alignment. We, therefore, performed mechanical tests following the direction of the nanofibers because this direction alone acts both as a carrier for cell adhesion and elongation, and simultaneously as a guide for the motor proteins and so ensures that the SKM is structurally organized and functioning.
Figure 1C: It is also interesting to discuss about the difference in the tenacity of each scaffold.
We thank the referee for the valuable suggestion. However, for the proposed use of these scaffolds, we focused on the mechanical characteristics related to limited values of strain, such as elastic modulus or strain at yield values. Tenacity, being related to the failure values of the scaffolds, was not investigated by us.
6. Line 262-264: I don't agree with this sentence. PLCL 70/30 seems to have a change in slope (from elastic to plastic) much sooner (about 2%). Please, verify this.
In agreement with the referee’s comment, upon further analysis, it was found that indeed the yield point of the PLCL 70/30 specimen can be better identified at 2.1 ± 0.6 % strain. Consequently, the elastic modulus of this sample was recalculated, reporting the value of 14.5 ± 2.5 MPa. We then corrected these values in both text and figures at Results section paragraph 3.1 and Figure 1D as follows:
“Stress-strain measurements of the PLCL scaffolds (Figure 1C) showed that an increase in the CL co-units content from 5 mol% to 15 mol% caused a slight decrease in the elastic modulus from 89MPa to 73MPa whereas the elastic modulus dropped down to 14.5MPa when the content of CL units increases up to 30 mol%. It is worth remarking that, among the studied samples, the PLCL 70/30 one best approximates the mechanical behavior of SKM tissue (Young modulus of muscle fibers in relaxed state 61 ± 5 kPa[36])”. (lines 279-288)
Lines 264-268: This sentence need more explanation and a reference that support it.
In agreement with the referee’s comment, we slightly modified the original sentence to explain it better, and we added a reference to support it. All other references were renumbered accordingly.
The original sentence was: “The results of the mechanical analysis can be explained considering the differences in the Tg values of the copolymers: PLCL 95/05 and 85/15 have a Tg higher than RT, and therefore they show a similar elastic modulus, whereas PLCL 70/30 has a lower Tg, around RT, and a corresponding lower stiffness.”
The sentence was modified as follows:
“The results of the mechanical analysis, and in particular the differences in the elastic modulus values of the samples, can be interpreted considering the solid-state properties of the copolymers [14]. Both PLCL 95/05 and PLCL 85/15 show a Tg higher than RT (57°C and 42°C respectively) and therefore they are both glassy materials in the conditions of the mechanical measurements, showing a similar elastic modulus. PLCL 70/30, on the other hand, has a Tg lower than that of the other two copolymers (22°C) that, being in the RT range, justifies the rubber-like behavior and the corresponding lower stiffness of PLCL 70/30.” (lines 292-299)
Line 272: The alignament of the sample could be measured by FIJI.
We thank the referee for the comment. As suggested, sample alignment was measured with FIJI's directionality plugin. The measurements showed that 68% of the nanofibers are oriented within the range 0-18° from the direction of rotation of the collector (circumferential direction), while only 2.6% within the range 72-90°. The manuscript was updated both in the paragraph 2.3 at the Materials and Methods section and 3.1 at the Results section as follows:
“The fiber orientation of the mats of aligned nanofibers was calculated using the Directionality plugin of Fiji [32], employing the Local Gradient Orientation method, following a previous validated procedure [33].” (lines 153-155)
“SEM analysis revealed the presence of regular and bead-free fibers, while the analysis with Fiji's Directionality plugin showed fibers well aligned with the direction of rotation of the collector (Figure 1E left image). 68% of the total amount of nanofibers resulted aligned within a range of 0-18° from the direction of rotation while only 2.6% within the range 72-90°)”. (lines 302-307)
Lines 278-285: How much collagen was deposited in the sample? Where is the collagen: integrated into the fibers or in the holes between them? Is the collagen distributed evenly throughout the sample? Does the sample dry between this immersion and the deposition of the cells? I believe that this method for incorporating collagen into the samples is interesting, and the revision has implemented some new tests to support it, but there are still many doubts to be resolved in it that can condition the biological behavior of the samples. If these questions have been solved in other work. This must be referenced.
We thank the referee for the comment. Through gravimetric measurements of uncoated and coated PLCL mat (by means of an analytical balance), the collagen coated PLCL mat showed an increase in weight of 0.25 mg ± SD 0.07, suggesting that the increased weight is due to collagen deposition. Coating occurred on the entire surface of the mat as proved by WCA and FTIR-ATR analysis. As described in the Materials and Methods section in paragraph 2.5, after sterilization the mat was coated O.N with the collagen solution and before cell seeding was washed with PBS and not subjected to any drying passage.
Sincerely yours,
Prof.ssa Giovanna Cenacchi, MD
Lab Patologia e Diagnostica Subcellulare, DIBINEM

Round 2
Reviewer 1 Report
I accept manuscript in the present form.
Author Response
Bologna, 31/01/2023
Manuscript ID number: bioengineering-2148991
Type of manuscript: Article
Title: Synthetic electrospun poly(L-lactide-co- ε-caprolactone) scaffold for skeletal muscle myogenesis: molecular and morphological analysis on C2C12 cell line
Dear Reviewer,
thank you for your revision.
Sincerely yours,
Prof.ssa Giovanna Cenacchi, MD
Lab Patologia e Diagnostica Subcellulare, DIBINEM

Reviewer 2 Report
The revised manuscript entitled “Synthetic electrospun poly(L-lactide-co- ε-caprolactone) scaffold for skeletal muscle myogenesis: molecular and morphological analysis on C2C12 cell line “ provides very detailed information based on my previous recommendations point-by-point clearly. However, the title itself is somehow unclear/misleading since the substrate components and fabrication method are not new discoveries. The title should more clearly indicate the major aims/novelty based on their responses and current present form it does not.
Author Response
Manuscript ID number: bioengineering-2148991
Type of manuscript: Article
Title: Synthetic electrospun poly(L-lactide-co- ε-caprolactone) scaffold for skeletal muscle myogenesis: molecular and morphological analysis on C2C12 cell line
Dear Reviewer,
thank for your comments. Please find below our response with regards to the question raised.
Reviewer #2: The revised manuscript entitled “Synthetic electrospun poly(L-lactide-co- ε-caprolactone) scaffold for skeletal muscle myogenesis: molecular and morphological analysis on C2C12 cell line “ provides very detailed information based on my previous recommendations point-by-point clearly. However, the title itself is somehow unclear/misleading since the substrate components and fabrication method are not new discoveries. The title should more clearly indicate the major aims/novelty based on their responses and current present form it does not.
We thank the referee and we agree with his conclusion; we modified the manuscript title as follows:
“Electrospun poly(L-lactide-co- ε-caprolactone) scaffold potentiates C2C12 myoblast bioactivity and acts as a stimulus for cell commitment in skeletal muscle myogenesis”.
Sincerely yours,
Prof.ssa Giovanna Cenacchi, MD
Lab Patologia e Diagnostica Subcellulare, DIBINEM

Reviewer 3 Report
This work present a molecular and morphological analysis of synthetuc electrospun scaffolds that are interesting right now in the engineering sector (specifically in tissue engineering). In this way, this work is novel, interesting and suitable for its publication in this journal. Nevertheless, some comments must be solved before its publication:
-Section 2.1: Is it possible to supply the molecular weight of the copolymers? This weight is decisive in its electrospinning process.
-Line 126-133: Why those electrospinning parameters were selected? An explanation of their selection or a bibliographic reference must be included.
- Why the used funtionalization method is better than incorporate the collagen directly in the electrospun solution? (https://doi.org/10.1002/jbm.a.37156)
-Line 146: Mechanical tests were performed in the nanofibers directions. What about the perpendicular direction? This direction will also support stresses and its resistance will be much less. It is important to consider it also in these tests (https://doi.org/10.1007/s10704-020-00460-4).
-Figure 1C: It is also interesting to discuss about the difference in the tenacity of each scaffold.
-Line 262-264: I don't agree with this sentence. PLCL 70/30 seems to have a change in slope (from elastic to plastic) much sooner (about 2%). Please, verify this.
-Lines 264-268: This sentence need more explanation and a reference that support it.
-Line 272: The alignament of the sample could be measured by FIJI.
-Lines 278-285: How much collagen was deposited in the sample? Where is the collagen: integrated into the fibers or in the holes between them? Is the collagen distributed evenly throughout the sample? Does the sample dry between this immersion and the deposition of the cells? I believe that this method for incorporating collagen into the samples is interesting, and the revision has implemented some new tests to support it, but there are still many doubts to be resolved in it that can condition the biological behavior of the samples. If these questions have been solved in other work. This must be referenced.
Author Response

(The authors gave the same response as above.)
